# Serum Triglycerides and Atherosclerotic Cardiovascular Disease: Insights from Clinical and Genetic Studies

**DOI:** 10.3390/nu10111789

**Published:** 2018-11-17

**Authors:** Hayato Tada, Atsushi Nohara, Masa-aki Kawashiri

**Affiliations:** Division of Cardiovascular and Internal Medicine, Kanazawa University Graduate School of Medical Sciences, 13-1 Takara-machi, Kanazawa 920-8641, Japan; a-nohara@med.kanazawa-u.ac.jp (A.N.); mk@med.kanazawa-u.ac.jp (M.-a.K.)

**Keywords:** remnant lipoproteins, triglycerides, cholesterol, genetics

## Abstract

Lipoproteins are a major risk factor for atherosclerotic cardiovascular diseases (ASCVD). Among the lipoproteins, low-density lipoproteins (LDL) have been shown to be causally associated with ASCVD development. In contrast, triglycerides or triglyceride-rich lipoproteins receive less attention than LDL because there is little definite evidence from randomized controlled trials. A Mendelian randomization study has recently been published in which a causal association could be estimated with observational datasets. Using such Mendelian randomization studies, ranging from common to rare genetic variations, triglycerides seem to be causally associated with ASCVD outcomes independent of LDL. Although the “causal association” of serum triglycerides and ASCVD is difficult to assert, accumulated evidence from clinical and Mendelian randomization studies, using common and rare genetic variations, strongly supports such an association. In this article, we provide a summary of investigations focusing on important causal associations between serum triglycerides and ASCVD from the clinical point of view.

## 1. Introduction

Low-density lipoprotein (LDL) cholesterol has been shown as a causal risk factor for atherosclerotic cardiovascular diseases (ASCVD). Over the decades, LDL-lowering therapies including statins, ezetimibe, and proprotein convertase subtilisin-kexin type 9 (PCSK9) inhibitors, have been introduced in clinical settings and have contributed to better prognosis [1,2,3]. In addition to LDL cholesterol, triglyceride-rich lipoproteins are also considered important causal risk factors for ASCVD [4], and are usually considered one of the major “residual” risk factors of standard LDL-lowering therapies [5]. However, triglycerides or triglyceride-rich lipoproteins are not attracting sufficient attention from researchers and physicians, particularly from cardiologists who should carefully assess patient risk [6]. This may be because there are only a few randomized controlled trials (RCTs) supporting causal associations between triglycerides and ASCVD outcomes. In this article, we provide evidence from various approaches, including clinical data and common to rare variations in human genetics, supporting the conclusion that serum triglycerides are a causal residual risk factor for ASCVD.

## 2. What Are Triglycerides?

A triglyceride consists of glycerol and three fatty acids [7]. In the liver, triglyceride hydrolysis provides fatty acids for β-oxidation, signaling, and substrates for the assembly of very low-density lipoprotein (VLDL) triglycerides. Triglycerides cannot penetrate into cell membranes. A special enzyme located on the blood vessel walls called lipoprotein lipase (LPL) catabolizes those triglycerides to free fatty acids and glycerol. Triglycerides are not present in the blood per se because they are hydrophobic. Triglycerides are one of the major components of apolipoproteins, such as chylomicron, VLDL, intermediate-density lipoprotein (IDL), LDL, and high-density lipoprotein (HDL) (Figure 1). Accordingly, we need to consider which apolipoproteins are increased or decreased and which apolipoproteins are associated with human diseases.

## 3. Triglycerides and ASCVD

Triglyceride-rich lipoproteins are larger than LDL, and their penetration into arterial walls has been presumed to be limited to biophysical considerations alone. However, both apoB100 and apoB48 can be extracted from atherosclerotic plaque [8]. Accumulating experimental evidence suggests that VLDL can penetrate into the vessel intima, contributing to the development of atherosclerosis, whereas chylomicron and chylomicron remnants are too large to penetrate into the endothelial layer. Moreover, the triglyceride-rich lipoproteins do not need to undergo oxidative modification to be incorporated into macrophages because the macrophages recognize apolipoprotein E on the lipoproteins’ surface, triggering lipoprotein uptake [9,10]. It is therefore biologically plausible that triglyceride-rich lipoproteins are atherogenic. Furthermore, triglycerides are significantly associated with risk of ASCVD even in patients with familial hypercholesterolemia (FH) mainly caused by LDL receptor dysfunction, where LDL cholesterol is critically increasing their risk for ASCVD [11].

In addition to those clinical data, a useful scheme can illustrate a causal association between factors (typically a biomarker) and outcomes. In contrast to randomized controlled trials (RCTs), which need extensive time and effort, the Mendelian randomization study utilizes certain genotypes as instruments to assess a causal association between biomarkers and outcomes (Figure 2) [12]. This approach can be considered as a proxy of an RCT, wherein we can assume that the confounding variables could be randomized evenly. Therefore, this interesting approach can be considered a natural RCT. In the case of triglycerides, recent Mendelian randomization studies show causal relationships with ASCVD. Using common genetic variations, Do et al., conducted an interesting study creating different 12 logistic models to determine whether triglycerides causally influence risk for ASCVD. They adjusted the effects of LDL cholesterol and/or HDL cholesterol levels on ASCVD risk in the models, and found that the triglyceride level conferred by single nucleotide polymorphisms associated with triglyceride levels was significantly associated with risk for ASCVD [13]. Moreover, they also showed robust associations between rare genetic variations associated with triglycerides and ASCVD risk in targeted and exome-wide analyses [14,15]. Interestingly, the gene-based association testing indicated that rare genetic variations associated with lower triglycerides were consistently associated with reduced risk for ASCVD [16]. These facts, ranging from clinical data and genetic studies with common as well as rare variants, collectively suggest that triglycerides are causally associated with ASCVD.

## 4. Lessons from Extreme Cases (Severe Hypertriglyceridemia)

With regard to the association between triglycerides and ASCVD, researchers, including ourselves, have reported that significantly elevated triglyceride level (>1000 mg/dL) is not always associated with ASCVD [17]. In addition, ASCVD is not frequently observed in cases with lipoprotein lipase (LPL) deficiency, with triglyceride levels typically significantly elevated [18], although heterozygous mutation carrier has been associated with ASCVD [14].

## 5. Interventions for Hypertriglyceridemia

Body weight control and dietary modification are effective treatments for hypertriglyceridemia [19,20]. This may include restricting intake of carbohydrate, alcohol and omega-3 fatty acid [21]. Medications could be considered for patients with elevated triglyceride levels that do not respond to lifestyle changes. Fibrates could be considered first, based on RCTs, including the Helsinki Heart Study (HHS), Bezafibrate Infarction Prevention (BIP), Veterans Affairs high-density lipoprotein intervention trial, Fenofibrate Intervention and Event Lowering in Diabetes (FIELD), the Action to Control Cardiovascular Risk in Diabetes (ACCORD), and Diabetes Atherosclerosis Intervention Study (DAIS) showing beneficial effects in patients with elevated triglycerides and decreased HDL cholesterol levels [22,23,24,25,26,27]. Moreover, a novel peroxisome proliferator-activated receptor α (PPARα) modulator (SPPARMα) called pemafibrate showed higher potency and selectivity for activation of PPARα compared with fenofibrate [28]. This new drug is associated with fewer adverse effects and further reduction of triglyceride levels [29]. On the other hand, beneficial effect of omega-3 polyunsaturated fatty acids (n-3 PUFAs) to prevent ASCVD has been controversial. Although accumulated data from RCTs have not provided definitive evidence to reduce ASCVD risk using n-3 PUFA, many studies so far had several problems, including low dosage of n-3 PUFA and no assessment of triglycerides [30]. Upcoming trials using larger dosages of n-3 PUFA, and targeting patients with higher risk could illuminate the efficacy of this type of drug in ASCVD risk management.

Another way to reduce triglyceride is through apolipoprotein-C3 (APOC3) inhibition using anti-sense techniques. APOC3 plays a pivotal role in triglyceride-rich lipoprotein metabolism. Based on the findings from a Mendelian randomization study investigating the association between loss-of-function mutation in the APOC3 gene and ASCVD, APOC3 inhibition seems to reduce triglycerides and coronary heart disease events [31]. It has been shown that anti-sense oligonucleotides could reduce triglycerides in patients with hypertriglyceridemia by ~70% [32].

## 6. Fasting State or Post-prandial (Non-fasting) State?

Usually, serum triglyceride is assessed at fasting state. However, those assessments may not reflect average serum triglyceride levels and the associated risk for ASCVD. Measuring triglycerides in the post-prandial state has several advantages over fasting measurements. People in general are in almost always in a post-prandial state, which allows blood sampling without the need for fasting. On the other hand, if the Friedewald equation is used, LDL cholesterol may be underestimated with the presence of chylomicrons [33]. In addition, many RCTs used fasting triglyceride, and they provided evidence for risk assessment to prevent ASCVD. However, other epidemiological studies and major RCTs with statins used triglyceride levels assessed at post-prandial state, leading us to change our clinical practice using non-fasting blood samples [34,35,36,37]. In addition, in most of the Mendelian randomization and association studies focusing on the association between triglycerides and ASCVD outcomes, triglyceride levels were mainly measured in a non-fasting state [13,14,15,31]. Therefore, the European Atherosclerosis Society and European Federation of Clinical Chemistry and Laboratory Medicine (EAS/EFLM)’s joint consensus statement recommended that non-fasting blood samples should be routinely used for plasma lipid profile assessments [38]. However, we also recommend assessments in the fasting state because subjects may have some genetic or secondary causes of hypertriglyceridemia [17,18]. Moreover, it has been shown that post-prandial increase of triglycerides or remnant lipoproteins are one of the major causes for the development of ASCVD [39]. In addition, we have shown that the difference in such changes in post-prandial increase of triglycerides or remnant lipoproteins could be a factor for the phenotypic difference in similar diseases, such as FH, sitosterolemia, and autosomal recessive hypercholesterolemia (ARH) [40,41]. Accordingly, checking the post-prandial increase of such lipoproteins could be useful when assessing their ASCVD risk in detail, including in extreme situations.

## 7. Residual Risk Factor for ASCVD

Many researchers have been investigating for “residual” risk factors since the beneficial effect of statins was established [42]. Among these, triglycerides are associated with residual ASCVD events under the situation of adequate statin therapies (Figure 3) [5,43]. Furthermore, the studies showing such associations between triglycerides and residual ASCVD were high quality RCTs with a sufficient sample size. Only a few biomarkers, including triglycerides, have a good supporting “evidence” with high quality.

## 8. LPL Pathway and ASCVD

LPL has been shown to be one of the major drivers of catabolism of triglyceride-rich lipoproteins, including remnant lipoproteins [44]. However, determining the causal association between the LPL pathway and ASCVD development has been difficult. Recently, Kathiresan et al., investigated whether LPL pathway molecules were causally associated with ASCVD in large Mendelian randomization studies. As stated before, they showed that individuals with loss-of-function mutations in APOC3 gene exhibited lower odds for ASCVD incidents associated with lower triglycerides [31]. In addition, they found that apolipoprotein A5 missense mutations were significantly associated with early-onset of myocardial infarction [45]. Moreover, they found that rare variations in LPL gene were significantly associated with ASCVD [14]. Findings where LPL pathway molecules are consistently associated with ASCVD suggest a strong causal association between LPL pathway and ASCVD development (Figure 4).

## 9. Conclusion

In this review, we repeatedly emphasized that triglycerides are residual, modifiable, and causal ASCVD risk factors. More attention should be paid to triglycerides, in addition to LDL cholesterol, to reduce ASCVD events further. 

## Figures and Tables

**Figure 1 nutrients-10-01789-f001:**
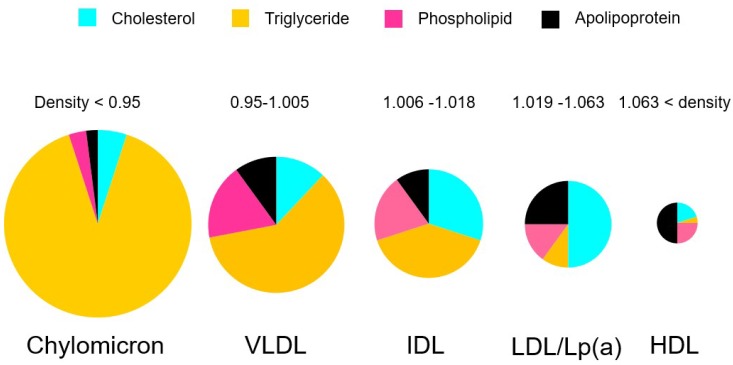
Lipid content of each lipoprotein. Light blue indicates cholesterol. Orange indicates triglycerides. Pink indicates phospholipid. Black indicates apolipoprotein. Chylomicron, very low-density lipoprotein (VLDL), and intermediate-density lipoprotein (IDL) are recognized as triglyceride-rich lipoproteins, and LDL is regarded as cholesterol lipoprotein.

**Figure 2 nutrients-10-01789-f002:**
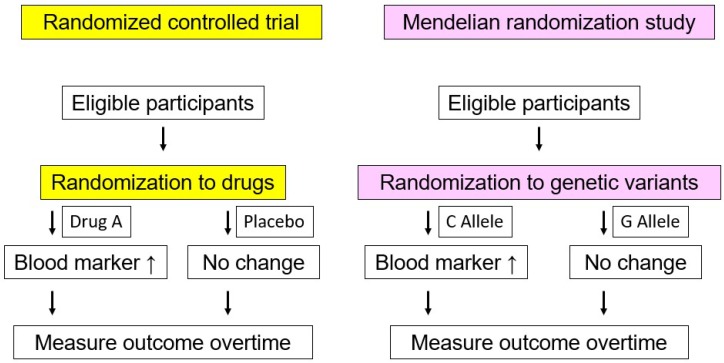
Scheme of randomized controlled trial and Mendelian randomization studies. The left panel indicates a randomized controlled trial. The right panel indicates a Mendelian randomized study. In a randomized controlled trial, participants are typically randomized either to placebo or actual drug, then we measure blood markers and associated outcomes. In this case, we can assume that intervention using particular drug associated with changes in blood marker lead to the changes in outcomes. On the other hand, in a Mendelian randomized study, participants are randomized to particular alleles, then we measure blood markers and associated outcomes. In this case, we can regard it as a proxy of a randomized controlled trial.

**Figure 3 nutrients-10-01789-f003:**
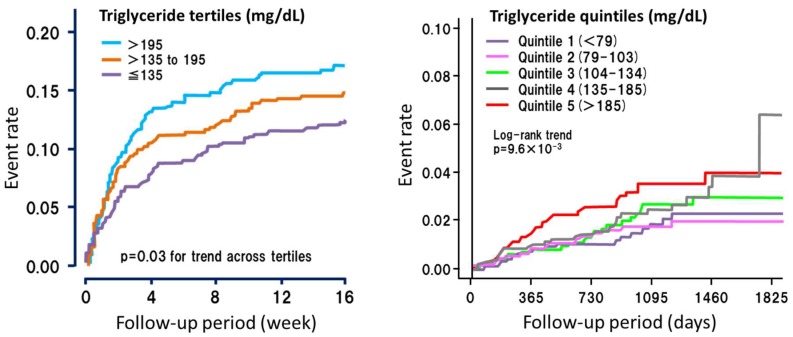
Cardiac events and triglycerides under statin therapies. The left panel shows short-term events after acute coronary syndrome (The Myocardial Ischemia Reduction with Aggressive Cholesterol Lowering: MIRACLE study). The subjects were divided according to tertiles of baseline triglyceride levels. The right panel shows the long-term events in the primary prevention group (standard versus intEnsive statin therapy for hypercholesteroleMic Patients with diAbetic retinopathy: EMPATHY study). The subjects were divided according to quintiles of baseline triglyceride levels. Both results strongly suggest that triglyceride is a residual risk factor under sufficient statin therapy for the development of ASCVD.

**Figure 4 nutrients-10-01789-f004:**
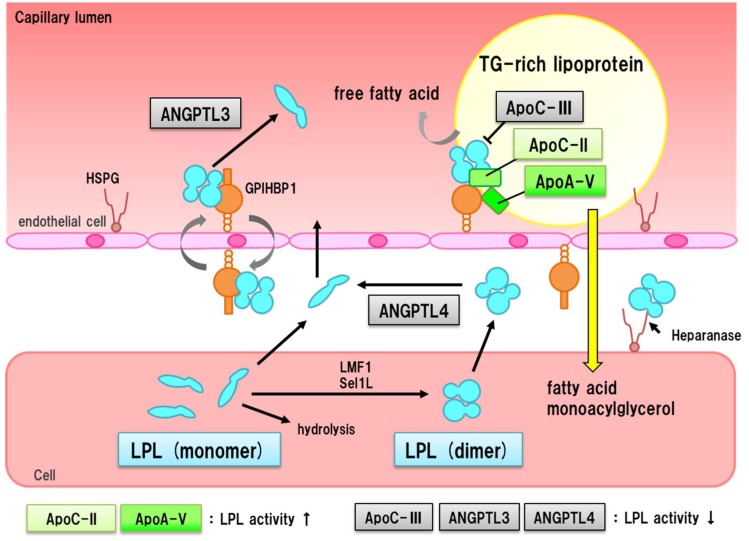
LPL pathway and associated proteins. Major proteins associated with LPL pathway are illustrated in this scheme. LPL is transported to endothelial cells, and binds to GPIHBP1. LPL hydrolyzes triglycerides on lipoproteins. APOC3, angiopoietin-like protein 3 (ANGPTL3), and angiopoietin-like protein 4 (ANGPTL4) play as inhibitor of LPL. On the other hand, APOC2 and APOA5 activate LPL. The fact that those proteins are significantly associated with ASCVD strongly suggests that LPL pathway is associated with the development of ASCVD.

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
