# Peer review of "Serum Triglycerides and Atherosclerotic Cardiovascular Disease: Insights from Clinical and Genetic Studies"

_nutrients, 2018, doi:10.3390/nu10111789_

Round 1
Reviewer 1 Report
This review is written very well. It reflects the actual opinion with respect to the role of triglycerides in the atherogenesis.
Some more important intervention studies (Helsinki Heart Study, FIELD, ACCORD, DAIS) should have been mentioned.
Minor comments
Page 1 Line 37: This may be because there are no randomized controlled trials (RCTs) supporting causal associations between triglycerides and ASCVD outcomes. – the FIELD and the ACCORD studies represent randomized studies showing an effect of a fibrate in patients with elevated triglycerides and low HDL-C
Page 2 Line 44: They are also present in the blood to enable bidirectional transference of adipose fat and blood glucose from the liver – triglycerides are not involved in the transport of glucose
Page 2 Line 65: Accumulating experimental evidence suggests that triglyceride-rich lipoproteins – “triglyceride-rich lipoproteins” should be replaced by VLDL
Page 3 Line 93: Figure 2. Schema of randomized controlled – Scheme
Table 1 – probably Reference 16 – no background and no explanation of these data is given; the interpretation in the text is not clearly reflected in the Table
Page 5 Line 133: If this Friedewald equation is used – the Friedewald equation
Paragraph 6: the authors should discuss the role of tests checking the postprandial increase of triglycerides
Page 6 Line 160: LPL is one of the major drivers of triglyceride-rich lipoproteins – drivers of the catabolism of triglyceride-rich lipoproteins
Reference 1: 1. Trialists C.T.. – wrong authors
Page 9 Line 331: of atherogenic dyslipidaemiaCardiovasc Diabetol – spelling error
Reviewer 2 Report
The author summarized the recently findings which supported the causal association between serum triglycerides and atherosclerotic cardiovascular disease (ASCVD) by clinical and Mendelian randomization studies.
1. Page 2 line 72, FH should be defined when it first time show up in the text.
2. It is better to have the description/explanation (legend) for Table 2 and Figure 4.
3. It is better to provide more description for every figure and table in the text, not just one sentence conclusion. It is difficult to understand how you got this conclusion from the present figure and table.
